# Dependence of Water-Permeable Chitosan Membranes on Chitosan Molecular Weight and Alkali Treatment

**DOI:** 10.3390/membranes10110351

**Published:** 2020-11-18

**Authors:** Ryo-ichi Nakayama, Koki Katsumata, Yuta Niwa, Norikazu Namiki

**Affiliations:** Department of Environmental Chemistry & Chemical Engineering, School of Advanced Engineering, Kogakuin University, 2665-1 Nakano-machi, Hachioji, Tokyo 192-0015, Japan; s316020@ns.kogakuin.ac.jp (K.K.); st13562@g.kogakuin.jp (Y.N.); nnamiki@cc.kogakuin.ac.jp (N.N.)

**Keywords:** chitosan, membrane, water content, water permeability, alkali treatment

## Abstract

Chitosan membranes were prepared by the casting method combined with alkali treatment. The molecular weight of chitosan and the alkali treatment influenced the water content and water permeability of the chitosan membranes. The water content increased as the NaOH concentration was increased from 1 to 5 mol/L. The water permeation flux of chitosan membranes with three different molecular weights increased linearly with the operating pressure and was highest for the membrane formed from chitosan with the lowest molecular weight. Membranes with a lower water content had a higher water flux. The membranes blocked 100% of compounds with molecular weights above methyl orange (MW = 327 Da). At 60 ≤ MW ≤ 600, the blocking rate strongly depended on the substance. The results confirmed that the membranes are suitable for compound separation, such as in purification and wastewater treatment.

## 1. Introduction

Chitin and chitosan are biopolymers contained in the exoskeletons of crustaceans that have recently attracted attention as reproducible biogenic components [1,2]. They are significant for effective resource utilization, because they can be obtained from shells that are discarded during the processing of crabs and shrimps for food products [3,4].

Chitin is formed from N-acetyl-D-glucosamine that is linked linearly with β-1,4 units, whereas chitosan is formed from D-glucosamine (i.e., the deacetylation product of chitin). Both structures are similar to cellulose [5,6].

Conventional industrial applications of chitosan include as a flocculent [7], adsorbent [8,9], and fiber [10], because it is commercially and continuously available at low cost. Chitosan is also anticipated to be a biocompatible material in functional gels for drug delivery systems [11,12,13] and as a membrane material for industrial separation tools [14,15,16]. Membranes offer several advantages over other separation techniques because of their low energy consumption, bulk production at continuous operation, and production of bio-products that are not thermally denatured. Industrial applications of membrane separation are wide-ranging and include fruit-juice condensation [17], artificial dialysis [18], desalination of seawater [19], and wastewater treatment [20,21,22].

A membrane is characterized by its mechanical strength (i.e., stress–strain relationship) and mass-transfer characteristics. The mechanical strength determines the handling fatigue life of the membrane in a module. The mass-transfer characteristics determine the molecular diffusion rate through the membrane, which is the main rate-limiting step of the separation process. In general, chitosan is dissolved in aqueous acetic acid [23]. To form a water-insoluble chitosan membrane, the acetic acid must be neutralized by basic components such as sodium hydroxide (NaOH). During preparation, the type and concentration of the basic aqueous solution is known to influence the coagulation rate and structure of the chitosan gel. Moreover, the deacetylation degree of chitosan affects the distilled water permeation characteristics of the membrane [24]. The film-forming properties of chitosan are affected by the molecular weight of the chitosan and the alkali treatment at the time of membrane formation. The latter is essential to stabilizing the film formation against dissolution in water.

In this study, the water content, mechanical strength, water permeability, and mass-transfer characteristics of chitosan membranes were regulated by controlling the molecular weight of chitosan and the alkali treatment of the casting solution.

## 2. Materials and Methods

### 2.1. Materials

Powders of chitosan with three different molecular weights were purchased from Sigma-Aldrich (St Louis, MO, USA). Table 1 lists the mean molecular weights of the chitosan powders, which were determined from the measured viscosity. The guaranteed viscosity range of chitosan in the database was based on special-grade sodium hydroxide. Acetic acid and other chemicals were purchased from Fujifilm Wako Pure Chemical Industries, Ltd. (Osaka, Japan).

### 2.2. Preparation of Chitosan Membrane

Figure 1 shows the procedure for preparing the chitosan membranes. Chitosan was dissolved in 1.7 mol/L of acetic acid solution (20 g/L). The chitosan solution (20 g) was dispensed into a petri dish (diameter of 7.5 cm) and then dried for 12 h at 333 K in a thermostatic chamber. Subsequently, the chitosan was gelled by immersion in NaOH at a sufficient concentration (volume of = 25 mL, NaOH concentration = 1–5 mol/L). The chitosan in the petri dish was continuously immersed in the NaOH solution for 15–360 min. The resulting membrane was washed with distilled water. After the alkali treatment, the swollen membrane easily separated from the glass plate; it was thoroughly washed with distilled water to remove any excess NaOH. The neutralized state was checked by immersing pH paper in the wash water.

### 2.3. Scanning Electron Microscopy

The membranes were snap-frozen in liquid nitrogen and then dried in a vacuum freeze dryer (RLE-103, Kyowa Vacuum Engineering. Co., Ltd., Tokyo, Japan) at 298 K for 24 h. Then, the dried membranes were sputter-coated with a thin Pt membrane, using a sputter-coater (E-1010 Ion Sputter, Hitachi, Ltd., Tokyo, Japan). Finally, cross-sectional images of the membranes were obtained using a scanning electron microscope (SEM) (Miniscope TM-1000, Hitachi, Ltd., Tokyo, Japan).

### 2.4. Water Content

To determine the internal structure of a swollen membrane, the volumetric water content (*H_V_*) was determined from the water content of the membrane. For this purpose, each membrane was cut into 4 × 4 cm squares. Because a membrane sequesters water in its void spaces, the volumetric water content reasonably approximates the void fraction of a membrane in the swollen state. Each membrane square was blotted with filter paper to remove the excess surface water and was then dried in a thermo-controlled oven (333 K, 24 h). The water loss was measured gravimetrically with an electronic balance (ER-180A; A&D Co. Ltd., Tokyo). The volumetric water content each membrane square was obtained by calculating its gravimetric change after swelling:(1)$$H_{V}=\frac{V_{w}}{V_{m}}$$
where *V_w_* is the volume of water in the membrane, and *V_m_* is the volume of the membrane.

### 2.5. Mechanical Strength

The mechanical strengths of the membranes were measured with a rheometer (CR-DX500, Sun Scientific Co., Ltd., Tokyo, Japan). The swollen membranes were cut into 1 × 4 cm samples and stretched at a rate of 1.0 mm s^−1^. The maximum stress and maximum strain were measured to characterize the mechanical strength. The maximum stress σ was calculated as follows:(2)$$\sigma =\frac{B_{max}}{A_{c}}$$
where *B_max_* is the maximum pre-breaking load, and *A_c_* is the cross-sectional area of the initial membrane. The maximum strain *λ* was calculated as follows:(3)$$\lambda =\frac{L_{0}-L_{i}}{L_{i}}\times 100$$
where *L_i_* and *L*_0_ are the membrane lengths in the initial and breaking states, respectively.

### 2.6. Water Permeability

The water permeability of the membrane was measured with an ultrafiltration apparatus (UHP-62K, Advantec Tokyo Kaisha, Ltd., Tokyo, Japan) [25]. Figure 2 presents a schematic of the module and the setup for testing the water permeation. The initial volume of the aqueous phase was 200 mL, and the effective membrane surface area was 2.13 × 10^−3^ m^2^. The operating pressure ΔP (50–200 kPa) was adjusted by introducing N_2_ gas at room temperature (298 K). A magnetic stirring bar was installed near the membrane surface and stirred at a constant speed of 3 s^−1^ in the aqueous phase. The mass of the permeated water was measured on an electric balance and was converted to the volumetric amount of permeated water according to the permeated water density. The volumetric water flux *J_v_* was then calculated as follows:(4)$$J_{V}=\frac{V_{p}}{A_{m}\cdot \ell \cdot t}$$
where *V_p_* is the volumetric amount of permeated water, *A_m_* is the membrane surface area, *ℓ* is the thickness of the swollen membrane, and *t* is the operating time. These tests were replicated three times. The results of the water permeability test of the membranes were presented with the associated standard deviation (±SD).

### 2.7. Measurement of the Mass Transfer Flux

A chitosan membrane prepared by the method described in Section 2.2 was installed in the ultrafiltration device, and 190 mⅬ of the sample solution was poured in the permeation cell. After the device was filled with the sample solution, it was sealed, and a vial was attached to the permeation outlet. The experiment was started by pressurizing the device to 100 kPa with N_2_ gas. The stirring speed was 190 min^−1^. Each experimental sample solution (urea (MW = 60 Da), D-glucose (MW = 180 Da), methyl orange (MW = 327 Da), and bordeaux S (MW = 604 Da)) was dissolved in water as a solvent. After the substance permeation experiment, the absorbance of the sample solution before and after permeation was measured with an extra-visible visible spectrophotometer (V-630IRM, JASCO). After the absorbance was measured, the concentrations *C_f_* and *C_p_* before and after permeation, respectively, were determined from the calibration curve of each sample solution. The apparent rejection rate *R* was then calculated as follows:(5)$$R=\frac{\left(C_{f}-C_{p}\right)}{C_{f}}\times 100$$

## 3. Results and Discussion

### 3.1. Scanning Electron Microscopy

Figure 3 shows scanning electron microscopy (SEM) images of the surfaces and cross-sections of the chitosan membranes prepared in solutions with various NaOH concentrations (C_NaOH_ = 1.0 and 5.0 mol/L) and crosslinking times (*t*_N_ = 15 and 180 min). The surfaces of chitosan membranes prepared in 1.0 mol/L NaOH were smooth, and more membrane formed with a longer crosslinking time. Meanwhile, the cross-section showed that the structure became denser with a longer crosslinking time. The chitosan membranes prepared in 5.0 mol/L NaOH developed a rough surface with a longer crosslinking time, and the membrane surface peeled off and deteriorated. Furthermore, the cross-section of the membrane showed voids in the internal structure with a shorter crosslinking time. Previous SEM images demonstrated a measurable change in the biopolymer networks induced by the alkali treatment [26,27].

Figure 4 shows SEM images of the membranes prepared from chitosans with different molecular weights (low, medium, and high) in 5 mol/L NaOH. All of the membranes had uniform and dense internal structures in the thickness direction, but the membrane prepared from chitosan with a high molecular weight developed voids through its cross-section.

### 3.2. Volumetric Water Content

Figure 5 shows the effect of the crosslinking time on the water contents of chitosan membranes prepared in 1.0 and 5.0 mol/L NaOH. For the chitosan membranes prepared in 1.0 mol/L NaOH, the water content decreased with increasing crosslinking time. This is probably because the network structure within the membrane densified as the crosslinking progressed. Conversely, the water content of the chitosan membranes prepared in 5.0 mol/L NaOH showed no significant change regardless of the crosslinking time. This was attributed to the rapid progression of the crosslinking in the concentrated NaOH aqueous solution, so the membrane was fully formed within a short time.

Figure 6 shows the effects of the chitosan molecular weight and NaOH concentration on the volume porosity. For all chitosan membranes, the volumetric water content increased with the NaOH concentration. This trend can be explained by the hydrogen bonds that crosslink the amino and hydroxyl groups of chitosan [28]. Chitosan polymer networks are principally crosslinked by hydrogen bonds between hydrogel groups and amino groups. Increasing the concentration of the basic aqueous solution is equivalent to increasing its ionic strength; thus, when the NaOH concentration was high, the ionic strength was also high and the hydrogen bonds were weakened. This may have increased the clearance between polymer chains owing to the weakened hydrogen bonds form the higher ionic strength of NaOH [29]. This suggests that the concentration of the basic aqueous solution contributes greatly to the surface and cross-sectional structures of chitosan membranes.

### 3.3. Mechanical Strength

Figure 7 and Figure 8 show the effect of the NaOH concentration on the maximum stress and maximum strain, respectively, at the time of membrane rupture. Increasing the NaOH concentration decreased the maximum breaking stress and the maximum strain. These trends might be explained by the decreased number of hydrogen bonds and weakening bonds between chitosan molecules as the NaOH concentration increased. The membrane prepared from chitosan with a high molecular weight exhibited greater mechanical strength than the other two membranes. This may be explained by the stronger crosslinking of its polymer chains, which contained many crosslinking points [30].

### 3.4. Water Permeability

Figure 9 shows the effect of the crosslinking time on the water flux through the chitosan membranes prepared in 1.0 and 5.0 mol/L NaOH. The water permeation flux decreased with increasing crosslinking time, regardless of the NaOH concentration. At longer crosslinking times, the interior of the membrane grew denser and suppressed the water flux. Lengthening the crosslinking time probably increased the tortuosity of the permeation pathway through the membrane.

Figure 10 shows the effect of the operating pressure on the water permeation flux through the membranes prepared from chitosan with different molecular weights. The water permeation flux increased linearly with the operating pressure for all membranes and was highest for the membrane formed from chitosan with a low molecular weight. This appears to be because the molecular chain length of the chitosan influences the water permeation pathway through the membrane.

Figure 11 plots the water flux as a function of porosity for the membranes prepared from different-molecular-weight chitosan in different NaOH concentrations. Increasing the molecular weight of chitosan increased the volumetric water content and decreased the water permeation flux of the membrane. In general, the water permeation path increased with porosity. These results suggest that many moisture regions were immobilized by the molecular chains in the cell structure of the chitosan membrane. These regions could not function as permeation pathways for water. However, in the membrane formed from chitosan with a low molecular weight, the volumetric water content decreased and the permeation flux of pure water increased. The superior water permeation performance of this sample can be explained by the molecular chain length of the chitosan.

### 3.5. Mass Permeation Performance of the Chitosan Membranes

Figure 12 plots the glucose inhibition rate a function of the crosslinking time for chitosan membranes prepared in 1.0 and 5.0 mol/L NaOH. Increasing the crosslinking time increased the glucose inhibition rate of the chitosan membranes in 1.0 mol/L NaOH. This trend can be explained by the reduced number of pores in the membrane as the crosslinking time elapsed; this blocked or narrowed the permeation channels to below the molecular size of glucose (8.7 Å). In contrast, the inhibition rate of membranes prepared in 5.0 mol/L NaOH did not change significantly after 120 min. Since the crosslinking in the membrane was rapidly completed during the formation process with excessive NaOH, the diameters of the mass permeation channels may have been robust against extended crosslinking times.

When molecules with high and low molecular weights are separated with a polymer, the membrane must block the target polymer and pass the smaller molecules. This property can be evaluated with the membrane fractionation performance. The fractional molecular weight of a membrane is defined as the molecular weight at which the apparent rejection is 90% or more. Since the separation by molecular weight is non-uniform, the fractional molecular weight covers a range of molecular weights. Thus, the molecular weight cutoff is an important performance index of biopolymer membranes and is a helpful guide for selecting a suitable membrane for a given purpose.

Figure 13 shows the fractional molecular curve of membranes prepared from chitosan with different molecular weights in 1.0 mol/L NaOH. The rejection rate increased with the molecular weight of the chitosan. The inhibition rate of particles with molecular weights above methyl orange (327 Da) was 100%. At 60 ≤ MW ≤ 600, the blocking rate changed remarkably with the molecular weight, which indicates that a fractional molecular weight was identified. This range includes the molecular weights of many functional food components such as amino acids, saccharides, and food polyphenols.

## 4. Conclusions

Membranes were successfully prepared from chitosan powder of different molecular weights of chitosan and with different alkali treatments. The volumetric water content of the chitosan membranes increased with the NaOH concentration regardless of the molecular weight of the chitosan. The membrane prepared from chitosan with a high molecular weight exhibited greater mechanical strength than the other membranes. The molecular weight and alkali treatment significantly affected the water permeation flux and mass transfer the prepared chitosan membranes. The water permeability was highest in the membrane prepared from chitosan with a low molecular weight. The water permeation flux increased 1.8-fold as the NaOH concentration was raised from 1.0 to 5.0 mol/L. The membranes had an inhibition rate of 100% for tested components with molecular weights above MW = 327 Da. At 60 ≤ MW ≤ 600, the blocking rates changed remarkably with the molecular weight, which indicates that a fractional molecular weight was identified.

In this study, the volumetric water flux increased with the NaOH concentration and molecular weight of the chitosan. The findings regarding the dominant role of the alkali treatment on both the physical properties and water permeability of chitosan membranes will help facilitate the production of chitosan membranes as a separation technology for water treatment and environment-compatible engineering.

## Figures and Tables

**Figure 1 membranes-10-00351-f001:**
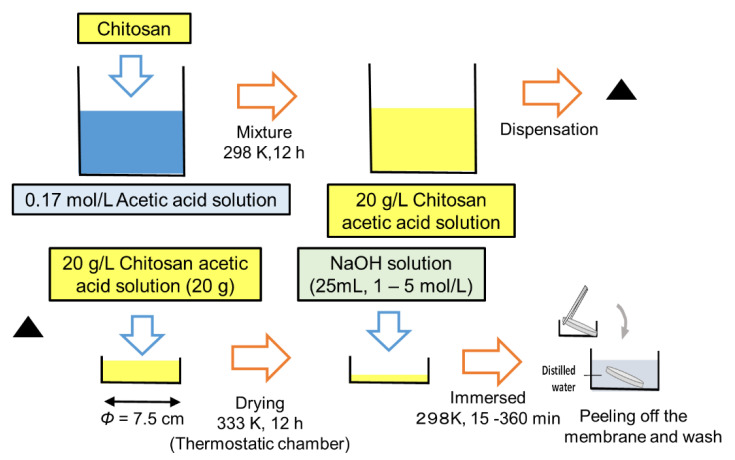
Experimental procedure for preparing the chitosan membranes.

**Figure 2 membranes-10-00351-f002:**
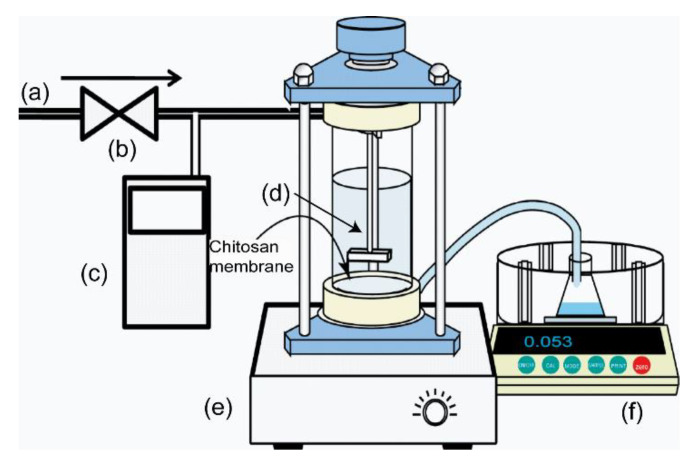
Schematic of the water permeation apparatus: (**a**) N_2_ gas inlet, (**b**) regulator, (**c**) transducer, (**d**) magnetic stirring bar, (**e**) magnetic stirrer, and (**f**) electronic balance.

**Figure 3 membranes-10-00351-f003:**
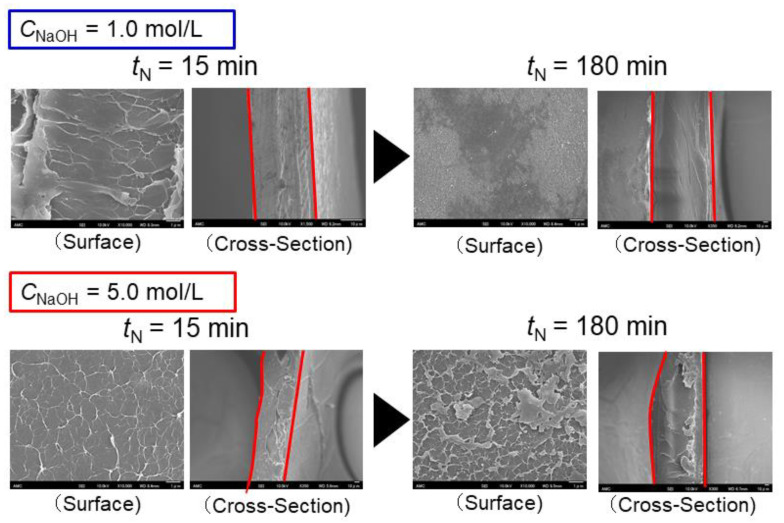
SEM images of the surfaces and cross-sections of chitosan membranes prepared in 1.0 mol/L NaOH (*t*_N_ = 15 and 180 min, upper panels) and 5.0 mol/L NaOH (*t*_N_ = 15 and 180 min, lower panels).

**Figure 4 membranes-10-00351-f004:**
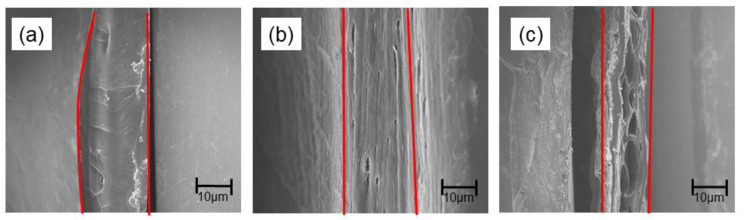
Cross-sectional SEM images of membranes prepared in 5 mol/L NaOH (*t*_N_ = 180 min) from chitosan of different molecular weights: (**a**) low, (**b**) medium, and (**c**) high.

**Figure 5 membranes-10-00351-f005:**
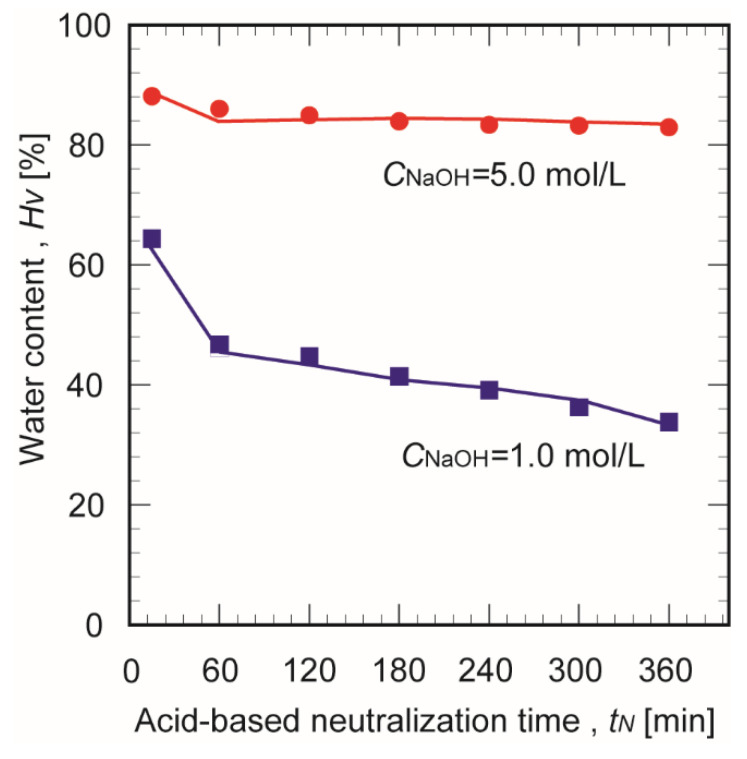
Effect of crosslinking time on the water content of chitosan membranes prepared in 1.0 and 5.0 mol/L NaOH.

**Figure 6 membranes-10-00351-f006:**
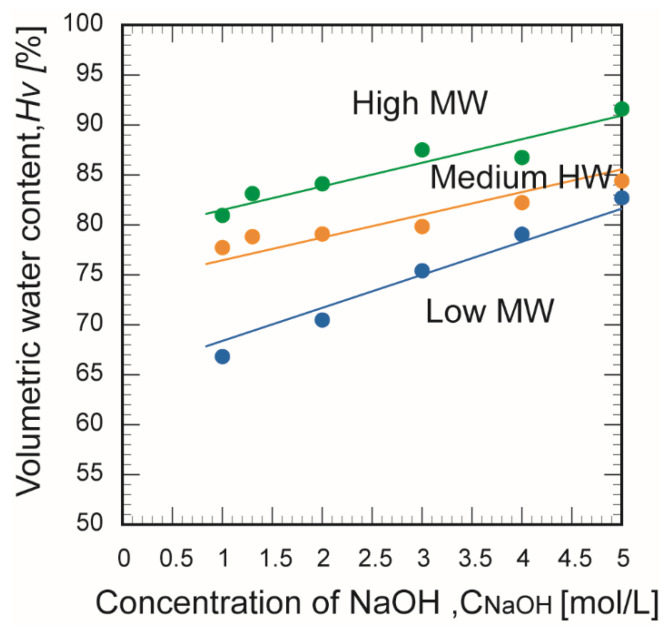
Effect of the chitosan molecular weight on the porosity and concentration of the aqueous NaOH solution during membrane formation.

**Figure 7 membranes-10-00351-f007:**
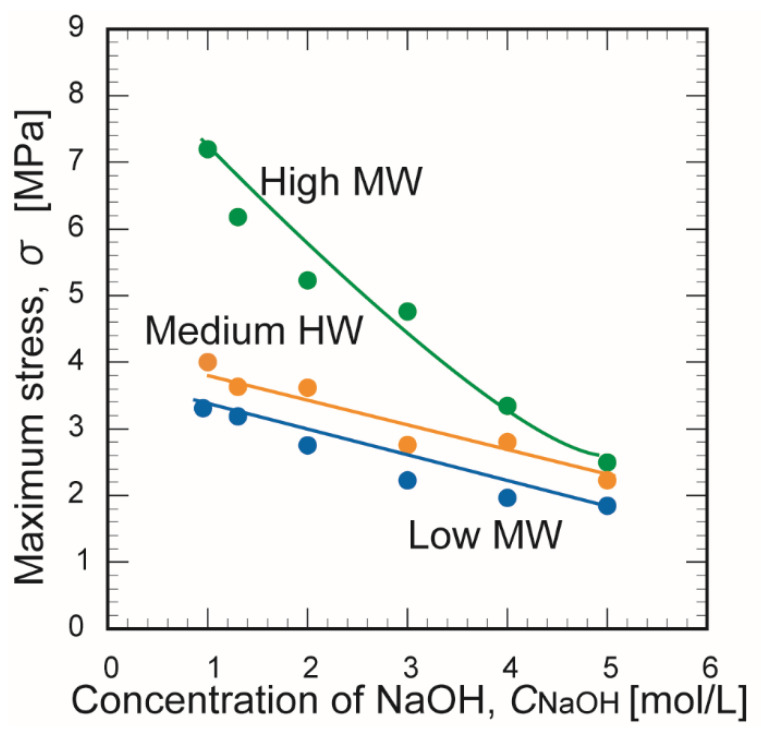
Effect of the NaOH concentration on the maximum stress of the three chitosan membranes at the time of rupture.

**Figure 8 membranes-10-00351-f008:**
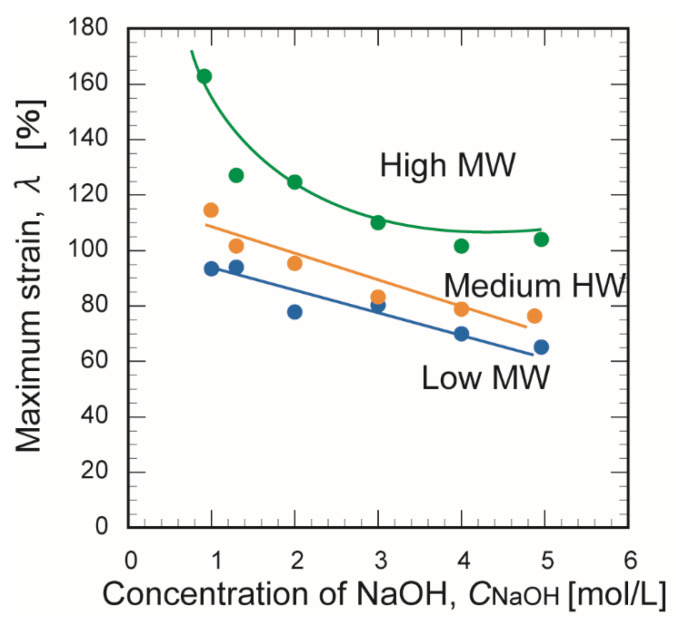
Effect of the NaOH concentration on the maximum strain of the three chitosan membranes at the time of rupture.

**Figure 9 membranes-10-00351-f009:**
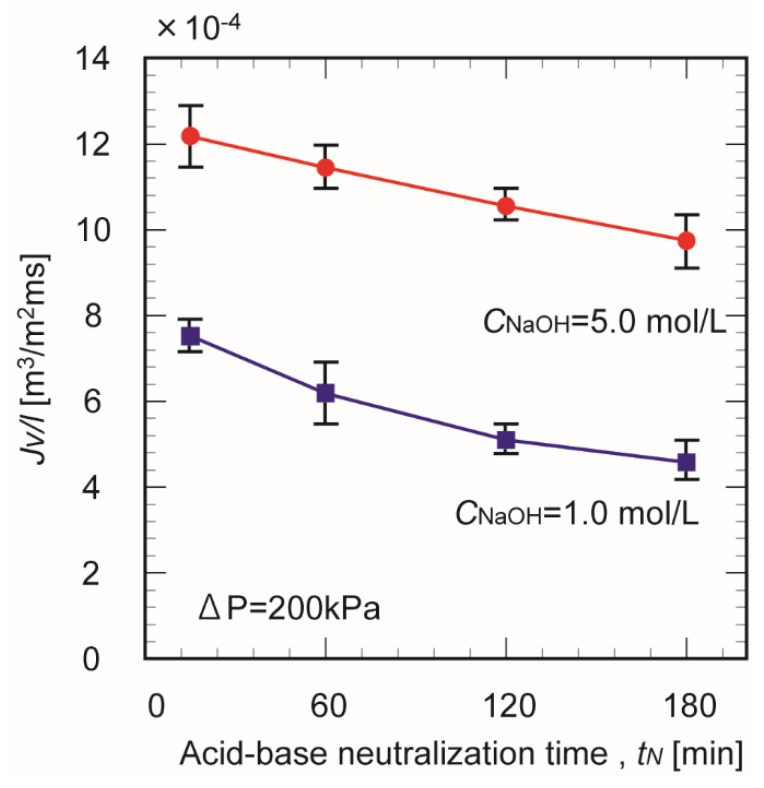
Effect of crosslinking time on the water permeation flux through chitosan membranes prepared in 1.0 and 5.0 mol/L NaOH.

**Figure 10 membranes-10-00351-f010:**
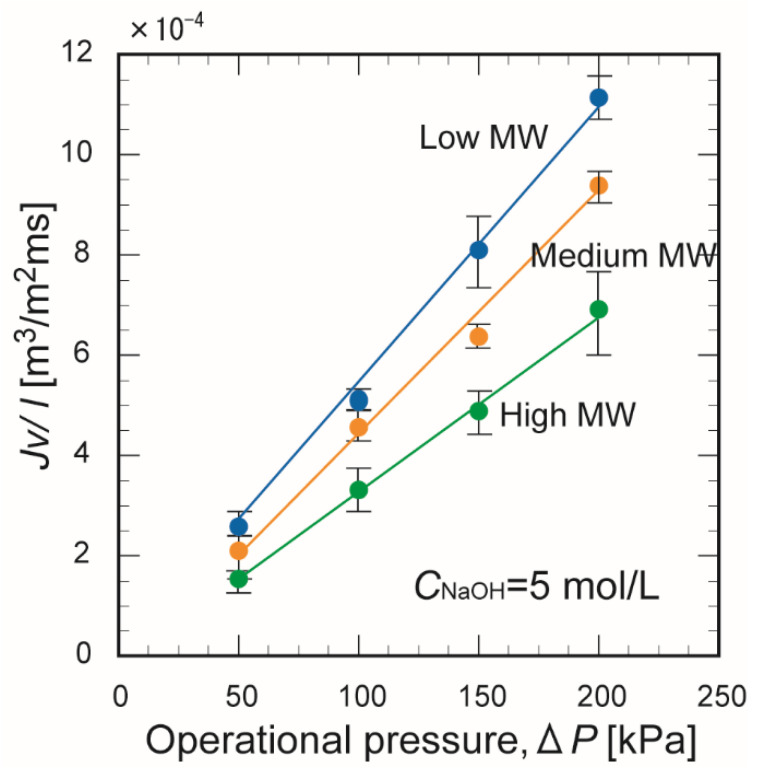
Effect of the operating pressure on the water flux through the chitosan membranes (*t*_N_ = 180 min).

**Figure 11 membranes-10-00351-f011:**
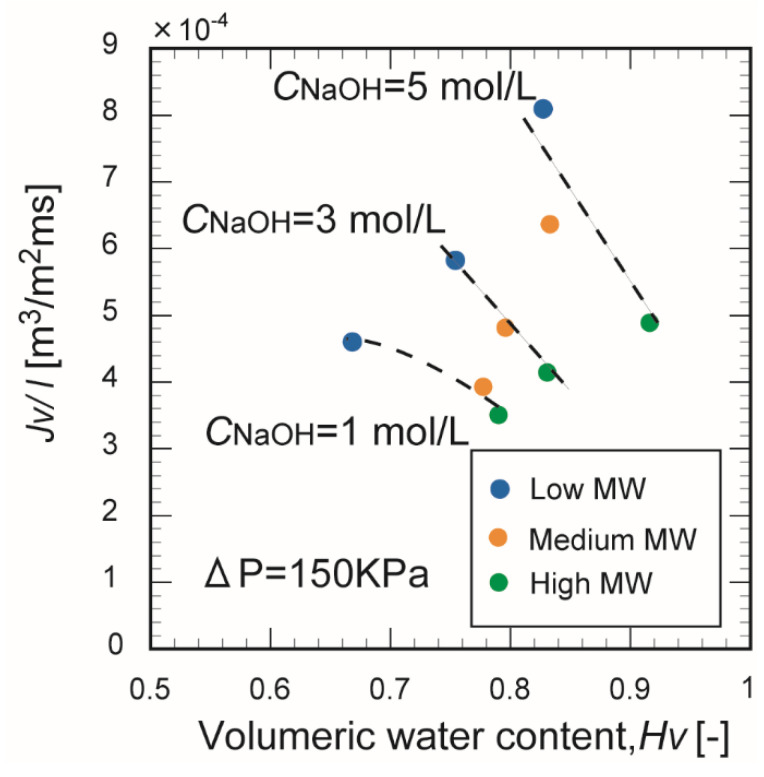
Correlation between the pure water flux and porosity of the chitosan membranes in different NaOH concentrations.

**Figure 12 membranes-10-00351-f012:**
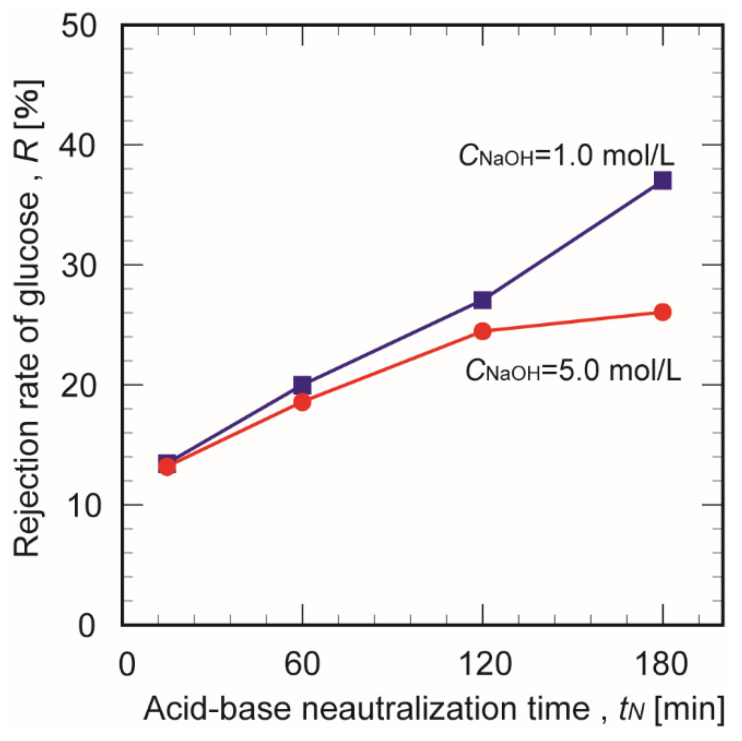
Effect of crosslinking time on the apparent rejection rate of the glucose ratio for chitosan membranes prepared in different NaOH concentrations (ΔP = 100 kPa).

**Figure 13 membranes-10-00351-f013:**
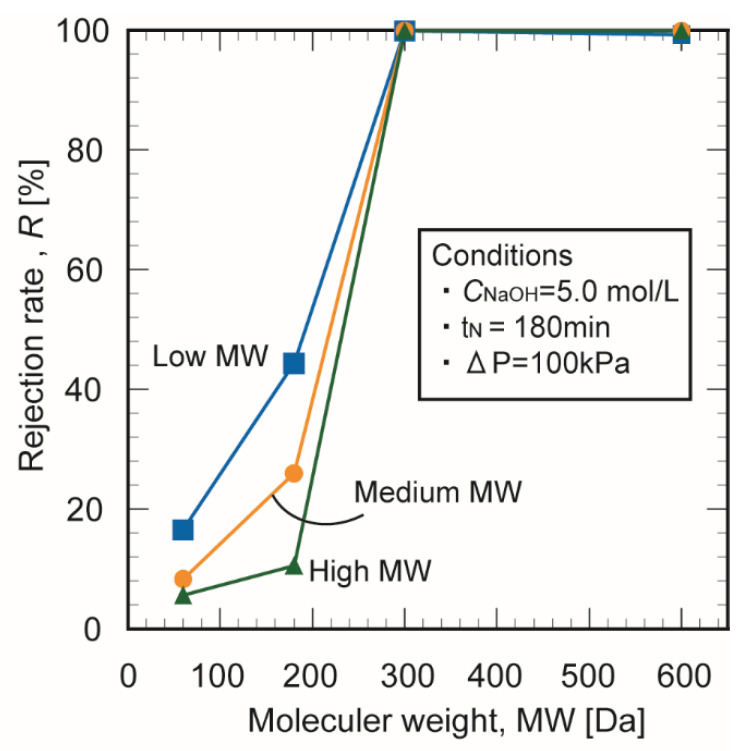
Fractional molecular curves of the chitosan membranes.

**Table 1 membranes-10-00351-t001:** Measured molecular weights of chitosan powders.

Type of Chitosan	Manufacture	Intrinsic Viscosity	Mean Molecular Weight, MW [Da] *	Guarantee Viscosity [cP] **
Chitosan (Low Viscosity)	Sigma-Aldrich	8.2	3.81 × 10^5^	20–200
Chitosan (Medium Viscosity)	Sigma-Aldrich	10.5	4.45 × 10^5^	200–800
Chitosan (High Viscosity)	Sigma-Aldrich	14.9	7.15 × 10^5^	800–2000

* Mean molecular weight of chitosan was determined by viscosity measurement. ** The range of viscosity was quoted from Sigma-Aldrich.

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
