# Peer review of "Dependence of Water-Permeable Chitosan Membranes on Chitosan Molecular Weight and Alkali Treatment"

_membranes, 2020, doi:10.3390/membranes10110351_

Round 1
Reviewer 1 Report
This manuscript reported dependence of water-permeable chitosan membranes on chitosan molecular weight and the acid-base neutralization process.
The followings should be considered before any further process.
- The concept of "acid-base neutralization" is confusing. Only the acid was neutralized in this study.
2. Explain the crosslinking between chitosan molecules in line 123. No functional group in chitosan can react in your experimental condition.
3. In line 154, "Increasing the concentration of the basic aqueous solution is equivalent to increasing its ionic strength, so when the NaOH concentration was high, the ionic strength was also high and the hydrogen bonds were weakened. Consequently, the crosslinks between the chitosan molecular chains were loosed and the distance between molecular chains indicated."
I cannot understand the latter sentence. Please re-write it.
And with this loosened chitiosan molecular chains, why the water permeability decreased with NaOH concentration? This seems controversial.
Author Response
Please find replies in the attachment.

Reviewer 2 Report
In this manuscript, through the casting method combined with the acid-base neutralization process, the chitosan membranes were synthesized. Then, the morphology of chitosan membranes were observed by scanning electron microscope. The water content, mechanical strength, water permeability and mass transfer flux were also discussed. Fiannly, by investigating the rejection rate of glucose, demonstrated that chitosan membranes can efficiently separte and purify some substance. In addition, I have some questions below: 1.In section 2.1, the viscosity of three different chitsan were measured, please list the manufacturer and name of the device.2. Please carefully check the grammar, there were too many minor issues:
L151: the effects of chitosan molecular weight NaOH concentration should be the effects of chitosan molecular weight and NaOH concentration;
L153: cross-limk should be cross-link;
L232: In the 60 ≤ MW ≤ 600 rante, rante should be range;
...
3. In figure 5, the title is maximum stress, but in the figure, it is maximum strain, please revise, the same as figure 6.
4. Draw a graphical to display the chitosan membranes synthesis schdule.
5. The SEM images of chitosan membrane should be also provided after filtration the glucose.
6. In L16, the membranes blocked 100% of compounds with molecular weights above methyl orange (MW = 600 Da), but in figure 11, the blocking rate can reach 100% when the MW > 300, please check and revise.
Author Response
Please find replies in the attachment.

Reviewer 3 Report
The manuscript ‘Dependence of water-permeable chitosan membranes on chitosan molecular weight and the acid-base neutralization process’ refers to a very interesting issue involving the prepare of a new type of membranes, but I believe that manuscript in present form, should not be published in the Membranes. It will be able to be published after considering the comments below:
The Introduction section should be corrected. Line 37, in recent years membranes are mainly used to reclamation of water from wastewater. That is why, I proposed implemented with some references concerning of wastewater treatment, e.g.:
A. Kowalik-Klimczak, E. Stanisławek, Reclamation of water from dairy wastewater using polymeric nanofiltration membranes, Desalination and Water Treatment 128 (2018) 364-371.
In the Materials and Methods section, the experimental installation should be described and the process parameters (pressure, feed cross flow velocity) used during membrane process will be added.
In the Results and Discussion section should be corrected.
Figures 8 and 9, How long did the filtration last? Was it repeated?
Figure 10, The effect of process pressure on rejection should be explained.
The stability of working is one of the most important factor to consider during preparation of new type of membranes. Please, explain what the stability of chitosan membranes during filtration of water. How is potential application of chitosan membranes?
Please update the literature. It is necessary to include the most recent literature reports in the manuscript.
Author Response
Please find replies in the attachment.

Round 2
Reviewer 1 Report
The authors made all the corrections as the reviewer suggested.
The reviewer agree to publish this manuscript.
Author Response
Thank you very much for your comments and affirmation.
Reviewer 2 Report
Chitin and chitosan, biopolymers contained in the exoskeletons of crustaceans, have recently 22 attracted attention as reproducible biogenic components. In this study, the authors employed a chitosan membranes, which was prepared by the casting method combined with the alkali treatment. It is true that the study could provide some basic and real information. However, some problems could be considered fully.
- Language usage needs to be revised.
- Conclusions should be revised.
- The references should be supplemented.
- How the application values of membranes developed in this study?
Author Response
Author’s reply to reviewer 2:
Thank you very much for your review of our manuscript. We found your comments very helpful, and they have enabled us to improve the paper again. We have read them carefully and revised the text accordingly.
Reviewer #2:
(Comment 1)
Language usage needs to be revised.
Our comment
The revised manuscript was carefully reviewed by an experienced editor whose first language is English and who specializes in editing papers written by scientists whose native language is not English again.
(Comment 2)
Conclusions should be revised.
Our comment:
We revised the minor issues mentioned above. Also the revised manuscript was carefully reviewed by an experienced editor whose first language is English and who specializes in editing papers written by scientists whose native language is not English.
(Comment 3)
The references should be supplemented.
Our comment:
Thank you for your comment. We have added the references.
(Comment 4)
How the application values of membranes developed in this study?
Our comment:
Thank you for your comment. In this study, the volumetric water flux increased with increasing NaOH concentration and chitosan molecular weight. Our findings regarding the dominant role of the alkali treatment both on the physical properties and water permeability of chitosan membranes will be anticipated as a benefit of producing chitosan membranes as separation technology for the water treatment and environment compatible engineering.
